# Comprehensive Functional Analysis of the bZIP Family in *Bletilla striata* Reveals That *BsbZIP13* Could Respond to Multiple Abiotic Stresses

**DOI:** 10.3390/ijms242015202

**Published:** 2023-10-15

**Authors:** Ru Zhou, Guangming Zhao, Siting Zheng, Siyuan Xie, Chan Lu, Shuai Liu, Zhezhi Wang, Junfeng Niu

**Affiliations:** 1National Engineering Laboratory for Resource Development of Endangered Crude Drugs in Northwest China, Xi’an 710119, China; zhouru0326@snnu.edu.cn (R.Z.); zhaoguangminghb@163.com (G.Z.); zhengsiting@snnu.edu.cn (S.Z.); xsy3094@snnu.edu.cn (S.X.); luchan@snnu.edu.cn (C.L.); liushuai28@snnu.edu.cn (S.L.); 2Key Laboratory of the Ministry of Education for Medicinal Resources and Natural Pharmaceutical Chemistry, Shaanxi Normal University, Xi’an 710119, China; 3College of Life Sciences, Shaanxi Normal University, Xi’an 710119, China

**Keywords:** abiotic stresses, *Bletilla striata*, bZIP TFs, expression pattern, protein interaction

## Abstract

Basic leucine zipper (bZIP) transcription factors (TFs) are one of the largest families involved in plant physiological processes such as biotic and abiotic responses, growth, and development, etc. In this study, 66 members of the bZIP family were identified in *Bletilla striata*, which were divided into 10 groups based on their phylogenetic relationships with *AtbZIPs*. A structural analysis of *BsbZIPs* revealed significant intron–exon differences among *BsbZIPs*. A total of 63 bZIP genes were distributed across 16 chromosomes in *B. striata*. The tissue-specific and germination stage expression patterns of *BsbZIPs* were based on RNA-seq. Stress-responsive expression analysis revealed that partial *BsbZIPs* were highly expressed under low temperatures, wounding, oxidative stress, and GA treatments. Furthermore, subcellular localization studies indicated that *BsbZIP13* was localized in the nucleus. Yeast two-hybrid (Y2H) and bimolecular fluorescence complementation (BiFC) assays suggested that *BsbZIP13* could interact with multiple *BsSnRK2s*. The results of this study provide insightful data regarding bZIP TF as one of the stress response regulators in *B. striata*, while providing a theoretical basis for transgenic and functional studies of the bZIP gene family in *B. striata*.

## 1. Introduction

Transcription factors (TFs) specifically regulate gene expression, and *bZIP* (basic leucine zipper) is one of the most abundant TF families in terms of its members in eukaryotes [1]. The bZIP gene family contains a highly conserved structural domain with a length of ~60–80 amino acids [2], which consists of a basic binding domain with a nuclear localization signal (NLS) and a binding motif to DNA (N-x7-R/K-x9) and the leucine zipper region. Since they possess a core DNA sequence (ACGT) that is tightly linked to the basic region [3,4], bZIP proteins bind to specific DNA sequences through a DNA-binding motif, which preferentially binds to the promoter G-box (CACGTG), C-box (GACGTC), A-box (TACGTA), and ABRE (CCACGTGG), etc. The leucine zipper dimer domain can form an amphiphilic α-helix structure that promotes bZIP protein dimerization and then binds to DNA [5,6]. Further, other specific structural domains exist in some plant bZIP proteins (e.g., proline and glutamine), which may interact with the transcriptional activation domain and participate in gene expression regulation [5].

It is known that bZIP TFs are involved in a variety of biological processes [7] (e.g., tissue and organ differentiation, embryogenesis, seed germination and maturation, plant senescence [8,9], photomorphogenesis [5], and protein storage), as well as the regulation of plant growth and developmental processes. The AtbZIP9/46 proteins play a role in the regulation of leaves as well as mid-vascular development [10]. Moreover, *OsABF1* is engaged in the regulation of the rice tassel stage and affects rice tasseling [11]. bZIP also regulates several secondary metabolite synthesis pathways in plants. *AabZIP1* was shown to activate the key enzyme genes, the artemisinin synthesis promoters *ADS* and *CYP71V1*, regulating artemisinin synthesis and inducing a significant increase in artemisinin levels [12]. Furthermore, bZIP is involved in biotic and abiotic stress responses and plays critical roles in the responses of plants to drought, cold, mechanical damage, and osmotic stress. *TabZIP6* reduces the frost resistance of Arabidopsis seedlings [13]. *OsbZIP71* is strongly induced by drought stress, and its overexpression significantly improves drought resistance in transgenic rice [14]. bZIP plays an important role in signaling cascades associated with the ABA pathway. Many bZIP TFs bind to ABA-responsive elements (ABRE) and regulate the transcription of these ABA-responsive genes. The *OsbZIP46* gene enhances the sensitivity of plants to ABA [15]. Other bZIP TFs factors negatively regulate ABA-mediated disease and stress resistance pathways. In rice lines that overexpress *OsbZIP52*, the expression of *OsLEA3*, *OsTPP1*, *Rab25*, and other abiotic stress-related genes are downregulated, which reduces tolerance to cold and drought in transgenic plants. This indicates that *OsbZIP52* is a negative regulator in response to cold and drought stress in rice [16].

*Bletilla striata* (Thunb. ex A. Murray) Rchb is a perennial herbaceous medicinal plant of the genus *Bletilla* in Orchidaceae. As a traditional Chinese herb, it has been used in China for more than 2000 years [17]. Preceding studies have shown that *B. striata* engages in various pharmacological antioxidant, anti-inflammatory, hemostatic healing, and immunoregulatory activities in vitro [18,19,20]. Medicinal plants are also affected by a variety of abiotic stresses during their growth, including drought, high salinity, and low temperatures, which will lead to reduced production [21,22]. Further, as a typical endosperm plant, it is difficult for *B. striata* to provide sufficient nutrients for seed germination due to its seeds being thin and lacking an endosperm, thus resulting in extremely challenging germination under natural conditions [20,23].

Currently, *B. striata* is gaining increasing popularity in the market due to its rich medicinal components and extensive range of pharmacological properties. However, the crop yields and productivity of *B. striata* can be severely hampered by abiotic stresses, including salinity, drought, and low temperatures. Thus, it is critical to develop strategies for the molecular breeding/genetic engineering of *B. striata* to mitigate rapidly changing climatic conditions. These approaches will require key genes to regulate abiotic stress response mechanisms in *B. striata*. Consequently, we performed genome-wide screening for the identification and expression analysis of *bZIP* TFs in the *B. striata* genome under abiotic stressors.

This study was based on the whole-genome database of *B. striata*, from which all its bZIP gene family members were screened and identified. Various bioinformatics analyses were performed, encompassing phylogenetic analysis, physicochemical properties, gene structure, conserved motifs, chromosomal localization, as well as synteny and *cis*-acting elements. Further, the expression patterns in different tissues under various abiotic stresses of *BsbZIPs* were analyzed in *B. striata* via RNA-seq. Finally, subcellular localization studies and yeast two-hybrid (Y2H) assays were performed. This study provides comprehensive data on BsbZIP TFs and their functional roles in abiotic stress responses in *B. striata*, while offering new clues toward resolving the difficult germination of endangered medicinal herbs of the Orchidaceae.

## 2. Results

### 2.1. Identification of bZIP TFs in B. striata

A total of 66 members of the *BsbZIP* gene family in *B. striata* were identified based on the genomic database. The results revealed that all *BsbZIPs* contained conserved structural domains (bZIP_1 (PF00170), bZIP_2 (PF07716), and bZIP_Maf (PF03131)). The characteristics of the *BsbZIPs* (Appendix A) were also analyzed, including the CDS length, protein molecular weight (Mw), isoelectric point (pI), aliphatic index, and the total average of hydropathicity grand average of hydropathy (GRAVY). The lengths of the BsbZIPs proteins ranged from 126 aa (*BsbZIP29/42*) to 664 aa (*BsbZIP7*); the isoelectric points ranged from 5.05 (*BsbZIP20*) to 11.65 (*BsbZIP15*); and the molecular weights ranged from 14.16 KDa (*BsbZIP8*) to 71.64 KDa (*BsbZIP7*). Subcellular localization predictions indicated that the *BsbZIPs* were localized in the nucleus.

Protein structure prediction revealed that the α-helix was the main component of BsbZIPs members (Appendix A). BsbZIPs proteins comprised non-transmembrane proteins without transmembrane structural domains, except for *BsbZIP7*/*24*/*27*/*35*/*36* (Appendix A). Further, all BsbZIPs had no signal peptides and were hydrophilic proteins (Appendix A). Phosphorylation site prediction indicated that BsbZIPs contained several phosphorylation sites, among which serine (S) was the most abundant, followed by threonine (T), and finally tyrosine (Y) (Appendix A, Appendix A). This suggested that the activities of BsbZIP proteins were potentially regulated by phosphorylation and dephosphorylation, and there may be additional co-regulated modalities.

### 2.2. Multiple Sequence Alignment and Phylogenetic Analysis of BsbZIP Protein

To elucidate the conserved characteristics of BsbZIPs domains, multiple sequence alignment analysis was performed based on protein sequences. Evidence from CDD, Pfam, and SMART validation indicated that 66 BsbZIPs proteins shared the bZIP structural domain, which was consistent with our prediction (Appendix A). The N-terminal basic amino acid region of the structural domain was highly conserved, which consisted of a basic DNA-binding region and an adjacent leucine zipper structure. The basic region contained an invariant N-X_7_-R/K motif, while the leucine zipper region consisted of leucine repeat heptapeptides or hydrophobic amino acid residues.

To further investigate the evolutionary relationships of *BsbZIPs*, a phylogenetic tree was developed, which included 141 members from *B. striata* and *Arabidopsis* (Figure 1). The results indicated that the bZIP gene family could be categorized into 10 subfamilies, referred to as A-H, I, and S. Clade S was the largest subfamily with 16 *BsbZIPs*, whereas clade D was the second subfamily that contained 11 *BsbZIPs*. Subfamily F was the smallest subclade that contained a single gene. Clades A, B, C, E, G, H, and I had 10, 2, 3, 3, 5, 3, and 10 members, respectively. It was also found that *BsbZIP13* and *AtbZIP39* were highly homologous in their evolutionary relationships, with both belonging to subfamily A. *AtbZIP39* was an ABI5 transcription factor, which was considered to be a key regulator of ABA regulation of seed germination [24], and it is speculated that *BsbZIP13* also has a similar function. *BsbZIP33*/*36* were not assigned to any subclade, which implied that large sequence differences and potentially functional changes occurred during evolution.

### 2.3. Analysis of Gene Structures and Conserved Motifs of BsbZIPs

To explore the structural features of *BsbZIPs*, we analyzed the introns and exons of each member (Figure 2C). The results showed that there were different numbers of exons in different subfamilies, while the same subfamily was relatively similar in the distribution and number of introns and exons. Fourteen genes had no introns, accounting for 21.2% of the total genes in the BsbZIP gene family. All members of subfamily A contained 1–3 introns, except for *BsbZIP6*, which had no introns. To investigate the characteristic regions of BsbZIP proteins, a total of 12 conserved motifs were predicted using the MEME online tool (Figure 2B and Appendix A). Different *BsbZIP* genes possessed various types of conserved motifs, which were similar in the same subfamily, and different numbers of motifs. Among the 12 different conserved motifs, motif 1 was present in almost all genes, whereas motif 2 existed only in subfamily D. Subfamily A contained motifs 1 and 9. Except for subfamily D, the motif composition of other *BsbZIP* subfamily members was relatively simple, with mostly 1–3 motifs. In conclusion, bZIP members in the same taxon shared similar conserved motif compositions and gene structures, which aligned with the results of phylogenetic analysis and strongly supported the reliability of taxon classification.

### 2.4. Analysis of Cis-acting Elements of BsbZIP Promoters

To explore the regulatory roles and potential functions of *BsbZIPs*, we performed a *cis*-acting element analysis based on their promoter regions (Figure 3). Following the removal of general transcriptional regulatory elements and functionally unknown elements, a variety of important *cis*-acting elements were revealed to be widely distributed in *BsbZIPs*. We set our focus on the response elements involved in plant growth and development, hormone regulation, and adversity. Among them, a total of 51 *BsbZIP* genes contained the TGACG-motif MeJa response element, 46 *BsbZIP* genes contained the light-responsive element (G-box), 29 and 12 *BsbZIP* genes contained low-temperature response elements (LTR) and ABRE elements in response to ABA signaling, respectively. The uneven distribution of *cis*-elements in the promoter region of *BsbZIP* genes may affect the expression of *BsbZIP* genes and lead to the functional differentiation of BsbZIP proteins.

### 2.5. Chromosomal Distribution and Synteny Analysis of BsbZIP Genes

We investigated the chromosomal distribution of 66 *BsbZIP* genes on 16 pseudochromosomes for *B. striata* (Figure 4A). Most of the *BsbZIPs* were abundant on chr6 (9 genes) and chr1 (8 genes), while there were only one each on chr7 and chr8. Seven of the *BsbZIP* genes were distributed on chr4, while both chr3 and 10 had five *BsbZIP* genes. Further, chr2, 5, and 15 contained four *BsbZIP* genes, respectively. From chromosomal localization analysis, it was found that the number of *BsbZIPs* on chr1, 4, and 6 were densely distributed, and the distribution positions of genes on other chromosomes were random.

To investigate the evolutionary relationships of *BsbZIPs* between species and the retention and loss of homologous genes, we created co-linearity plots for *B. striata* with *A. thaliana*, *Vitis vinifera*, and *Vanilla planifolia*, respectively (Figure 4B). There were 10, 30, and 45 pairs of homologous genes between *B. striata* and *A. thaliana*, *V. vinifera*, and *V. planifolia*, respectively. The results indicated that *B. striata* and *V. planifolia* possessed higher homologies.

### 2.6. Tissue Differential Expressions of BsbZIPs in B. striata

To discover the expression patterns of the bZIP gene family in different *B. striata* tissues, we analyzed the expressions of *bZIP* genes in roots, pseudobulbs, leaves, and flowers using transcriptomic data from RNA-Seq (Figure 5A). The results revealed that 31 *BsbZIP* genes were most highly expressed in roots, with the expression levels of 21 and 16 of the *BsbZIPs* exhibiting high expression levels in flowers and pseudobulbs, respectively, while only 11 *BsbZIPs* were highly expressed in leaves. Overall, the various expression levels of the *BsbZIP* genes in the four different tissues reflected their diverse and important roles in the growth and development of *B. striata*.

Furthermore, we analyzed the expression patterns of *bZIP* genes during different *B. striata* germination stages (Figure 5B). A total of 21 *BsbZIPs* genes, including *BsbZIP5*, *BsbZIP10*, and *BsbZIP13*, exhibited higher expression levels in BS0, which belonged to group A of the *BsbZIPs*. Only five genes (*BsbZIP24*, *BsbZIP30*, *BsbZIP34*, *BsbZIP40*, and *BsbZIP64*) exhibited significantly higher expression in BS1. Further, a total of 20 *BsbZIP* genes including *BsbZIP18* and *BsbZIP48* had the highest expression levels in BS5, while 9 *BsbZIP* genes including *BsbZIP8*, *BsbZIP30*, and *BsbZIP36* were more highly expressed in BS3. Moreover, the expression levels of *BsbZIP3*, *BsbZIP14*, *BsbZIP16*, and *BsbZIP53* were significantly higher during the BS3-BS5 stages. It was suggested that the expressions of different *BsbZIPs* may have been developmentally specific during seed germination. The expression patterns of the *BsbZIP* members in *B. striata* under different seed germination stages were beneficial toward understanding their detailed roles.

### 2.7. Expression Patterns of BsbZIPs in Response to Abiotic Stresses

To comprehensively evaluate the expression profiles in response to abiotic stress of the *BsbZIP* family, 12 members distributed in different subfamilies were randomly selected to undergo four types of stress treatments: hormone treatment (GA), oxidative stress (CuSO_4_), cold stress (low-temperature), and wounding. It was found that the response patterns of various genes to different treatments varied significantly (Figure 6). For the hormone treatment (GA), *BsbZIP20/31* was significantly downregulated, while other genes were upregulated. For the low-temperature treatment (4 °C), the expression levels of *BsbZIP15*/*26* were essentially unchanged after 3 h and upregulated after 6 h. *BsbZIP2*/*27*/*31* were significantly upregulated, with the expression of *BsbZIP27* being the most significant after 12 h, upregulated hundreds of times. Interestingly, under the heavy metal (CuSO_4_) treatment, *BsbZIP3/8* was initially downregulated, but then upregulated. The expression levels of *BsbZIP2*/*7*/*15*/*22*/*31* were significantly upregulated, with *BsbZIP27* being the most obviously upregulated at 12 h, by ~100 times. Under the wounding treatments, the most significant upregulation of *BsbZIP15* reached 16-fold at 3 h, followed by *BsbZIP24*, reaching 13-fold at 12 h. According to the evolutionary relationship, *BsbZIP13* was homologous to ABI5 (*ATbZIP39*), which was involved in the regulation of seed germination by the ABA signaling pathway [24]. To further understand whether *BsbZIP13* responds to ABA, it was subjected to ABA and drought treatment; this was mapped separately to highlight our subsequent focus on this gene. *BsbZIP13* was responsive to the stimuli of multiple abiotic stresses (Figure 7). The expression levels of *BsbZIP13* were significantly increased at all time points, as detected under the ABA and drought treatments, while this gene was significantly downregulated. Under the low-temperature and wounding treatments, its transcription levels were initially significantly decreased followed by substantial increases. Moreover, under the CuSO_4_ treatment, the *BsbZIP13* expression level was significantly higher at 12 h compared with the other treatment times. In conclusion, *BsbZIPs* played broad roles in abiotic stress tolerance mechanisms.

### 2.8. Localization of BsbZIP13 in the Nucleus

To reveal the potential functions of *BsbZIP13* in the transcriptional regulatory system, we created the fusion protein expression vectors p35s::BsbZIP13-eGFP, with the p35s::eGFP vector as a positive control group, and incorporated them into *Nicotiana tabacum* (tobacco), after which they were observed using laser confocal microscopy. The results revealed that the control treatment exhibited green fluorescence throughout the membrane and nucleus, while that of the GFP-BsbZIP13 fusion protein was specifically localized in the nucleus (Figure 8), which was consistent with the results of previous bioinformatics analyses.

### 2.9. BsbZIP13 Interactions with BsSnRK2.2/3/4/6

SnRK2 was a unique plant protein kinase that belongs to the SNF1/AMPK family, which was involved in phosphorylation activities and the ABA signaling pathway [25]. SnRK2 and ABI5 were both key factors in the ABA signaling pathway, so we wanted to know if there was an interaction between *BsbZIP13* and SnRK2s. Originally, pGBKT7-*BsbZIP13* (*BsbZIP13*-BD) vectors were developed, and a transactivation activity assay was performed in yeast. The results showed that *BsbZIP13*-BD-transformed yeast cells grew normally only on medium but not unchanged blue (Figure 9A). This indicated that *BsbZIP13* had no transcriptional activation activities. *BsSnRK2.2/3/4/5/6* were cloned into pGADT7 and Y2H-assayed for interactions between *BsbZIP13* and *BsSnRK2s*. The results revealed that *BsbZIP13* interacted with *BsSnRK2.2/3/4/6*, while there was no interactivity with *BsSnRK2.5* (Figure 9B). Finally, the BiFC assay was performed in onion epidermal cells for verification of the Y2H results, which were consistent with the Y2H results (Figure 10).

## 3. Discussion

bZIP TFs are extensively distributed and diverse in plants and play critical roles in plant development, seed maturation, stress responses, and pathogen defenses. To date, bZIP has been investigated for the model plant *A. thaliana* [26] but also on economic crops including rice [27], barley [28], and sorghum [29]. The research is focused on the regulatory mechanisms and functions under the stress-adverse conditions. As an important medicinal and ornamental plant [30], *B. striata* frequently encounters a variety of abiotic stresses during growth [21], including drought, high salinity, and low temperatures, which decreases its quality and yields. However, specific data on the bZIP gene family in *B. striata* are not known or have not been reported to date.

For this study, a total of 66 *BsbZIPs* were screened and identified based on the *B. striata* genomic database using various bioinformatics tools. Comparatively, other plants such as *A. thaliana* and *Pyrus bretschneideri* possessed 75 *AtbZIP* genes and 62 *PbbZIP* genes, respectively, and *Solanum lycopersicum* had 69 *SlbZIP* genes [3,31,32]. The number of *BsbZIPs* identified in this study was comparable to those reports, relatively smaller than *Zea mays* (125 members) [33] and *Sorghum bicolor* (92 members) [29], but larger than those of *Ricinus communis* (49 members) [34] and *V. vinifera* (55 members) [35]. Monocotyledon appeared to have a relatively larger bZIP family than Dicotyledon, which may have been due to the differentiation between Monocotyledons and Dicotyledons, as the number of evolved bZIP members in Monocotyledons was higher than that for Dicotyledons [29,34]. Multiple sequence alignment indicated that all members contained DNA-binding regions, N-X7-R/K motifs, and leucine zipper structures. Subsequently, based on clustering analysis and protein sequence similarities to *A. thaliana*, 66 *BsbZIPs* were clustered into 10 subfamilies (A-H, I, and S) [3]. These included the ABF subfamily (e.g., A subfamily in Figure 1), which was related to ABA and abiotic stress [36], TGA subfamily related to disease resistance (e.g., D subfamily in Figure 1) [37], GBF subfamily (e.g., G subfamily in Figure 1), and HY subfamily (e.g., H subfamily in Figure 1) related to light signal regulation [38]. Other species such as sesame (*Sesamum indicum*) [36], wild sweet potato (*Ipomoea trifida*) [37], and Tartary buckwheat (*Fagopyrum tataricum* Gaertn) [38] were classified into 9, 10, and 10 subfamilies, respectively, that were similar to those of the *BsbZIP* family classifications, which indicated the conservative evolutionary trend of the bZIP gene family.

Moreover, there were variations in the gene structures between different subfamilies, while the same subfamily members had similarities. For example, F and S subgroups had few introns, while D and G subfamilies contained large numbers of introns and exons. This was akin to licorice, where differences in gene structures may lead to functional differences between bZIP subfamilies [39]. In terms of gene structures, the intron numbers of subgroup A in different species did not vary. The subfamily A members of *B. striata* contained 1–3 introns, except for *BsbZIP6* (no intron), which was consistent with *Olea europaea* (1–5 introns) and watermelon (1–4 introns) [40,41]. The data above indicated that the gene structure of subgroup A was conserved during species evolution. Further, the results of interspecific collinearity analysis indicated that there were more collinearity gene pairs between *B. striata* and *V. planifolia* than between *B. striata* and *A. thaliana*, or between *B. striata* and *V. vinifera*. Species with more collinear gene pairs (*V. planifolia* and *B. striata*) appeared to have relatively closer evolutionary relationships. This was consistent with the collinearity results of other *B. striata* gene families [42]. 

The bZIP TFs are expressed in various tissues to mediate a range of developmental processes [43] and are essential for elucidating their functional roles. According to RNA-seq, about half the members of the *BsbZIP* family, including *BsbZIP10* and *BsbZIP18*, exhibited higher expression levels in roots, with similar results found for *Z. mays* [44]. It was shown that the overexpression of *ZmbZIP4* under normal conditions developed most robust primary roots and increased plant survival under abiotic stress. Interestingly, *ZmbZIP4* and *BsbZIP10/18* both belong to subfamily A, which indicated that these genes may have similar functions in the regulation of root growth and development. Further, we found that genes including *BsbZIP6/48/64* were highly expressed in pseudobulbs, which were unique adaptive stems in many orchidaceaes. Pseudobulbs have a strong capacity to retain water, which serve as drought buffers to maintain the water balance of plants [45]. Furthermore, approximately half of the *BsbZIP* family members exhibited high expression levels during BS3 and BS5, where the BS3 stage is a marker of the normal germination of orchidaceae seeds. This indicated that the *BsbZIP* family played a key role in the formation and development of *B. striata* protocorms and early seedling establishment [46]. Additionally, we found that the *BsbZIP* genes contained a series of *cis*-acting elements related to hormone responses and abiotic stresses, including ABA, GA (gibberellin), MeJA, wounding, light, drought, and low temperatures. It is well known that the function and regulation of genes are primarily determined by *cis*-regulatory elements [47]. Consequently, to gain insights into the functions of *BsbZIP* genes, we analyzed their expression levels in different subfamilies under various treatments (Figure 6). Under the CuSO_4_ treatment, *BsbZIP7/15/27/31* were significantly upregulated, and these genes may be considered potential candidates for investigating the role of BsbZIPs in response to oxidative stress. Under the low-temperature treatment, *BsbZIP15/20/27/31* were significantly upregulated. Furthermore, *BsbZIP20/31* contained the *cis*-acting element that responded to the low-temperature treatment, which indicated that these genes may be candidates for the investigation of BsBZIPs for low temperatures. It was shown that the SlAREB1 (a bZIP TF) regulates anthocyanin biosynthesis in *Solanum lycopersicum* seedlings under low temperatures via an ABA-dependent pathway [48]. The overexpression of *bZIP52* in *Oryza sativa* also increased sensitivity to cold [16]. Furthermore, the transient overexpression of BpChr04G00610 reduced the ability of birch to scavenge ROS under low-temperature stress, which suggested that it may be more sensitive to low temperatures [49]. GA is a tetracyclic diterpene growth factor that plays a critical role in regulating various aspects of plant development, such as seed germination, stem elongation, and flowering [50]. *OsABF1* repressed gibberellin biosynthesis to regulate plant height and seed germination in *Oryza sativa* L. [51]. *OsbZIP48* binds directly to the promoter of *OsKO2*, which encodes ent-kaurene oxidase 2 of the gibberellin biosynthetic pathway, to regulate GA homeostasis [52]. For hormone treatments with gibberellin (GA), the expression of *BsbZIP3/7* was most prominent at 12 h and exhibited a threefold upregulation. This gene may be considered a potential candidate for investigating the role of *BsbZIPs* in response to gibberellin. Here, we focused on detecting the changes in expression of *BsbZIP13* under GA, ABA, wounding, drought, CuSO_4_, and low-temperature treatments and found that *BsbZIP13* also responded to the stimuli of multiple hormone responses and abiotic stresses. In Arabidopsis, *AtbZIP39* has been shown to be an ABI5 transcription factor that synergistically regulates the ABRE-dependent ABA signaling involved in seed germination and post-germination growth [24]. In *B. striata*, its homologue was *BsbZIP13*, which was also shown in this study to be highly inducible to ABA treatments; thus, we predicted that it could be employed as a candidate gene for future functional research into seed germination.

For this study, subcellular localization studies revealed that *BsbZIP13* was localized in the nucleus, which was consistent with the TF characteristics and experimental studies of other organisms, such as *Moso bamboo* [53]. This observation suggested that the activation of these proteins may rely on post-translational modifications, or require interactions with unknown upstream factors [54]. SnRK2s are a key component of ABA signaling pathways mediated by ABA receptors (PYR/PYL/RCAR) and protein phosphatase 2Cs (PP2Cs), which play an essential role during plant growth, flowering, seed germination, and maturation [55,56]. ABI5 is a transcription factor that belongs to the bZIP gene family, which is a key regulator of abscisic acid (ABA)-mediated seed germination and plant growth [57]. In *A. thaliana*, research has shown that *SnRK2.2/3.6* regulate ABA-activated phosphorylation of AREB/ABF (ABI5) [58]. Interestingly, in a subsequent study, it was found that AREB/ABFs are phosphorylated by several protein kinases of the SnRK2 family [59]. To understand the interactions between BsbZIPs and BsSnRK2s, *BsbZIP13* was selected to perform Y2H and BiFC assays. The results indicated that *BsbZIP13* interacted with several *BsSnRK2s* genes, which was consistent with previous studies. Similarly, Previous studies have reported an interaction between *FtbZIP83* and *FtSnRK2.6/2.3* in *Tartary buckwheat* [60]. Taken together, these data suggest that the SnRK2-AREB/ABF signaling pathway plays a key role in cross-talking that is induced by abiotic stress signaling in plants. This also suggests that *BsbZIP13* is essential for the ABA signaling pathways. Subsequently, we will further verify the function of *BsbZIP13* in ABA regulation of seed germination through transgenic strains.

## 4. Materials and Methods

### 4.1. Genome-Wide Identification of bZIP Genes in B. striata

All *BsbZIPs* were obtained from the *B. striata* genomic database. The corresponding protein sequences of 75 *AtbZIP* TFs were retrieved from the TAIR database (https://www.arabidopsis.org/, accessed on 13 March 2023). The HMM (Hidden Markov Model) maps of the bZIP_1 structural domain (PF00170), bZIP_2 structural domain (PF07716), and bZIP_Maf structural domain (PF03131) were obtained from the Pfam protein family database (http://pfam.xfam.org/, accessed on 16 March 2023). Subsequently, further identification and screening were simultaneously performed using SMART (http://smart.embl.de/, accessed on 18 March 2023), CDD (https://www.ncbi.nlm.nih.gov/cdd/, accessed on 20 March 2023), and InterPro (http://www.ebi.ac.uk/interpro/, accessed on 22 March 2023) [61,62,63]. Several protein sequences without conserved domains were eliminated, and any TFs that contained the conserved domain were included as members of the *bZIP* gene family in *B. striata*. Finally, additional analysis was performed using these named bZIP sequences. The ExPASy ProtParam (https://www.expasy.org/, accessed on 24 March 2023) proteomics website and WoLF PSORT (https://wolfpsort.hgc.jp/, accessed on 26 March 2023) online tool were employed to predict the physicochemical characteristics and subcellular localization of BsbZIP proteins, respectively [64,65]. SOPMA (http://npsa-pbil.ibcp.fr/cgi-bin/npsa_automat.pl?page=npsa_sopma.html, accessed on 26 March 2023) and TMHMM v2.0 (http://www.cbs.dtu.dk/services/TMHMM/, accessed on 27 March 2023) were used to predict the secondary structures and transmembrane structural domains of the *BsbZIP* proteins [66,67]. SignalP v5.0 (https://services.healthtech.dtu.dk/services/SignalP-5.0/, accessed on 29 March 2023) was then utilized to predict the signal peptides of the *BsbZIP* proteins [68]. Finally, the phosphorylation of BsbZIP protein sites was predicted and analyzed with NetPhos v2.0 online software [69]. Default settings were used for the software tools above.

### 4.2. Multiple Sequence Alignment and Phylogenetic Analysis of BsbZIPs

Clustal v1.83 was employed for the multiple sequence alignment of BsbZIP proteins in *B. striata* under default settings [70]. All protein sequences of the conserved BsbZIP domains were obtained using SMART online software. The phylogenetic trees of bZIP proteins from *B. striata* and *A. thaliana* were developed using MEGA v7.0 (Mega Limited, Auckland, New Zealand) via the Neighbor-Joining (NJ) method.

### 4.3. Analysis of the Gene Structures, Conserved Motifs, and Cis-acting Elements of BsbZIPs

The exon–intron position data of 66 *BsbZIP* genes were obtained from the genomic database. All BsbZIP protein sequences were entered on the MEME online website with the maximum motif number set to 12, maximum and minimum widths of 50 and 10, respectively, and other default values [71]. The 2500 bp promoter sequence upstream of the *BsbZIP* gene was intercepted from the genomic database, and the *cis*-acting elements of the *BsbZIP* gene members were predicted and categorized using the PlantCARE database (http://bioinformatics.psb.ugent.be/webtools/plantcare/html, accessed on 28 March 2023) [72]. TBtools v1.089 (Chen, C., et al, China) was used to display the *BsbZIP* gene structures, conserved motifs, and *cis*-acting elements [73].

### 4.4. Synteny Analysis and Chromosomal Distribution of BsbZIP Genes

The homologies between *B. striata* and three other plants (*A. thaliana*, *V. vinifera*, and *V. planifolia*) were investigated using MCScanX (Wang, Y., et al, China) and visualized by TBtools v1.089. The *V. vinifera* and *V. planifolia* data were obtained from the NCBI database. The chromosomal location data for *BsbZIP* genes were obtained from the genome annotation information, which were graphically displayed using TBtools v1.089.

### 4.5. Plant Materials

The plants and seeds of *B. striata* were collected from the National Engineering Laboratory of Northwest Endangered Herb Resources Development, Shaanxi Normal University. The *B. striata* seedlings under similar growth conditions were transplanted into square plastic pots containing humus and sand (1:2) under the parameters of 30 °C, 55% relative humidity, and 12 h light/12 h dark.

Transcriptome sequencing was performed on four different tissues (roots, pseudobulbs, leaves, and flowers) of *B. striata* during flowering, and the tissue-specific expression patterns of *BsbZIP* genes were further elucidated. Full viable dried pods were selected, and a small number of seeds were extracted for sterilization and then sowed into a solid MS medium with NAA (1.0 mg/L) and 6-BA (2.0 mg/L) and incubated at 30 °C, 55% relative humidity, and 16 h light/8 h dark conditions. The seeds were collected at 0 days (BS0), 5 days (BS1), 10 days (BS2), 20 days (BS3), 28 days (BS4), and 35 days (BS5) of germination for transcriptome sequencing to determine the expression patterns at different germination stages of *B. striata* [74].

Subsequently, for two-month-old seedlings, the expression profiles related to different treatments were analyzed. For hormonal treatments, the surfaces of the aboveground portions of the seedlings were sprayed with 100 µM GA and ABA, respectively. For oxidative stress, the seedlings were sprayed with 100 µM CuSO_4_ [74]. The wounding treatment involved an incision (1 cm) that was made along the center of a seedling leaf. To quantify low-temperature responses, the plants were placed in an incubator at 4 °C, while for the drought treatment, the surfaces of the aboveground portions of the seedlings were sprayed with 20% PEG6000 [75]. The plants were collected under different stress conditions at four corresponding periods (1 h, 3 h, 6 h, and 12 h) with three replicates, where after all the samples were immediately frozen in liquid nitrogen and stored at −80℃ for further RNA isolation.

### 4.6. RNA Extraction and q-PCR Analysis

RNA was extracted from whole *B. striata* plants using the Polysaccharide and Polyphenol Plant Rapid RNA Isolation Kit (TaKaRa, Dalian, China), and cDNA was synthesized from 1 μg of total RNA according to the instructions of the HiScript II Q-Selective Reverse Transcriptase Kit (Vazyme Biotechnology, Nanjing, China). qPCR was performed using a LightCycler 96 system (Roche Diagnostics GmbH) with SYBR Green qPCR Master MIX (Vazyme, Nanjing, China). In our laboratory, three biological and three technical replicates were used for each reaction, and *BsGAPDH* was used as a common internal reference gene. The data obtained were used to calculate the relative expression levels of the *BsbZIP* gene via the 2^−ΔΔCT^ method [76]. Finally, all data were analyzed using GraphPad Prism 8.0 (San Diego, CA, USA) and Excel software (Microsoft, Redmond, MA, USA) and plotted with the help of TBtools v1.089. All qPCR primers used in this study are listed in Appendix A.

### 4.7. Subcellular Localization Analysis of BsbZIP13

We selected *BsbZIP13* for subcellular localization studies and initially cloned the full-length CDS sequence of *BsbZIP13* into the TOPO vector (Vazyme). This gene was then introduced into the pDONR207 vector (without a termination codon) using the BP recombination reaction from Gateway Technology (Invitrogen, Carlsbad, CA, USA). The primers are listed in Appendix A. Finally, it was cloned into the pEarleyGate103 vector by an LR recombination reaction to form pEarleyGate103-BsbZIP13. The recombinant plasmid was transformed to Agrobacterium EHA105, and the prepared suspensions were injected into tobacco (*N. tabacum*) leaves with a syringe and left for three days to observe the GFP fluorescence (488 nm excitation of GFP green fluorescence) using a high-resolution confocal laser microscope (Leica TCS SP5, LEICA, Wetzlar, Germany).

### 4.8. Protein Interaction Analysis

The full-length sequences of *BsbZIP13* (involving the stop codon) were first introduced into the pDONR207 vector using the BP recombination reaction from Gateway Technologies (Invitrogen, Carlsbad, CA, USA) and then introduced into the pGBKT7 vector using the LR recombination reaction. The primers are shown in Appendix A. The transformed clones were mated and selected on a synthetic dropout (SD) medium lacking tryptophan (Trp) and (SD/−Trp), or SD medium lacking Trp, histidine (His), and adenine (Ade) but containing X-alpha-gal (SD/−Trp/−His/−Ade/+X-alpha-gal), with the empty pGBKT7 vectors serving as negative controls.

The recombinant pGADT7-BsSnRK2s vectors were developed based on pDONR207-BsSnRK2s (with the inclusion of the termination codon) using the Gateway system. Here, we used the recombinant vector constructed in the laboratory for the experiment [74]. Following the co-transformation of yeast strain AH109 with pGBKT7-BsbZIP13, yeast cells were cultured on a double-selective medium (SD/−Leu/−Trp) at 28 °C for 3 d. Positive clones were then sequentially screened with X-alpha-gal on SD/−Ade/−His/−Leu/−Trp medium to detect the interactions between *BsbZIP13* and *BsSnRK2s*. pGBKT7-p53 and pGADT7 vector combinations were used as a positive control, while pGBKT7-lam and pGADT7 vector combinations were used as a negative control.

The pEarleyGate202-YC-BsbZIP13 was constructed based on pDONR207-BsbZIP13 without a stop codon. The primers are shown in Appendix A. pEarleyGate201-YN-BsSnRK2s recombinant vectors constructed in the laboratory were used for the experiment [74]. YN-BsSnRK2s and YC-BsbZIP13 recombinant plasmids were transformed into Agrobacterium EHA105, and the onion epidermis pre-grown for one day was put into the Agrobacterium immersion solution containing recombinant plasmid vector and empty plasmid, dark for 16h, observed and photographed under a fluorescence microscope. The combination of YC-BsbZIP13/YN and YN-BsSnRK2s/YC vectors were used separately as negative controls.

## 5. Conclusions

In conclusion, this study provided the first comprehensive data on bZIP TFs in *B. striata* through a genome-wide analysis. A total of 63 genes were localized on 16 pseudochromosomes, and a comprehensive bioinformatics analysis was conducted to elucidate the physicochemical properties, terms of phylogeny, genetic structures, conserved domains, *cis*-acting elements, and collinearity. The expression profiles in different tissues and germination stages were diverse and found potential candidates for studying different abiotic stresses. *BsbZIP13* responded significantly to a wide range of abiotic stresses, especially ABA treatment. Additionally, subcellular localization analysis revealed that *BsbZIP13* was localized in the nucleus. The results of Y2H and BiFC assays suggested that *BsbZIP13* could interact with *BsSnRK2s*. As a homolog of ABI5, *BsbZIP13* has been preliminarily shown to play a key role in ABA regulation of seed germination, but its function in transgenic lines of *B. striata* needs to be further verified.

## Figures and Tables

**Figure 1 ijms-24-15202-f001:**
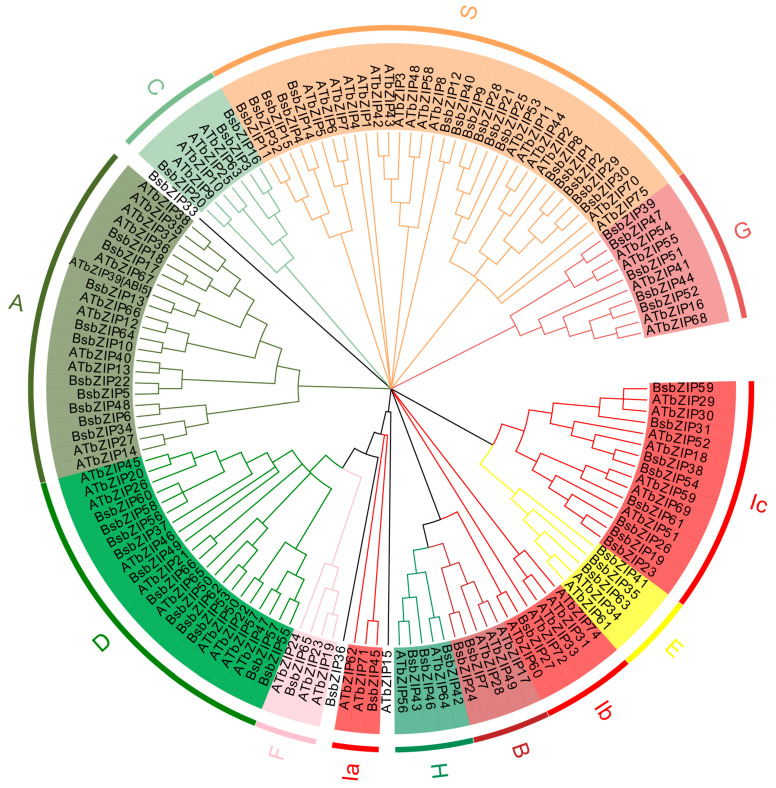
Comparative phylogenetic trees showed relationships between BsbZIP and AtbZIP domains. Using MEGA v7.0 software, the bZIPs sequences of *Bletilla striata* (66) and *Arabidopsis thaliana* (75) were bootstrapped with 1000 replicates to construct unrooted neighbor-joining (NJ) trees. The name of groups (A–I and S) are shown outside of the circle, indicating different bZIP subgroups. Various colors represent different subfamilies.

**Figure 2 ijms-24-15202-f002:**
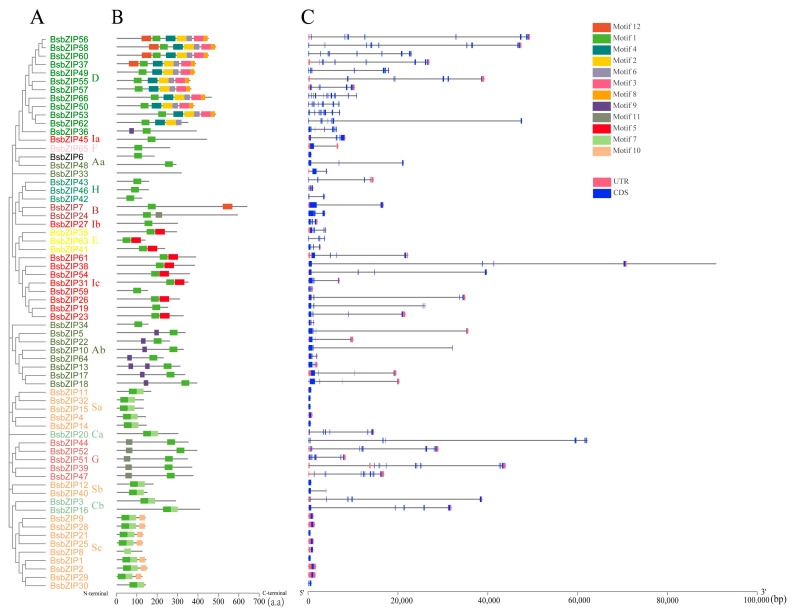
Phylogenetic tree, motifs, and gene structures in *BsbZIP* genes from *B. striata*. (**A**) The phylogenetic tree contained 66 bZIP proteins (*BsbZIP1* to *BsbZIP66*). Various colors represent different subfamilies. (**B**) Motif analysis of BsbZIP proteins. Colored boxes numbered 1–12 indicate different patterns. The length of the proteins can be estimated by the scale at the bottom. (**C**) Exon–intron structures of *BsbZIP* genes.

**Figure 3 ijms-24-15202-f003:**
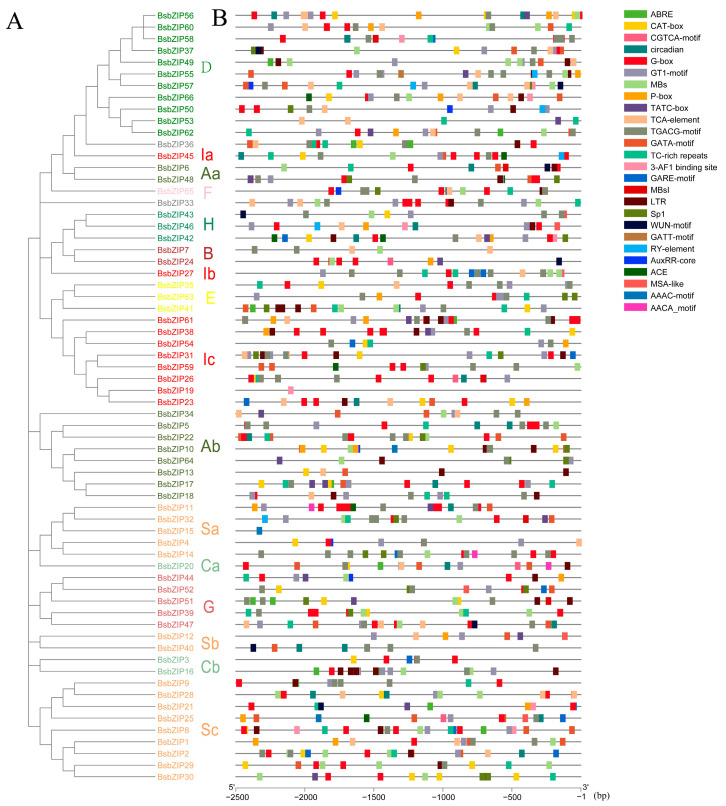
Distribution of various *cis*-acting elements in the promoter region of *B. striata bZIP* genes. (**A**) Phylogenetic relationship analysis of BsbZIP proteins. Different leaf background colors indicate the different subgroups (A–I and S). (**B**) Predicted *cis*-acting elements in the bZIP genes associated with hormonal and abiotic stresses. The length of the upstream proteins may be inferred from the scale at the bottom.

**Figure 4 ijms-24-15202-f004:**
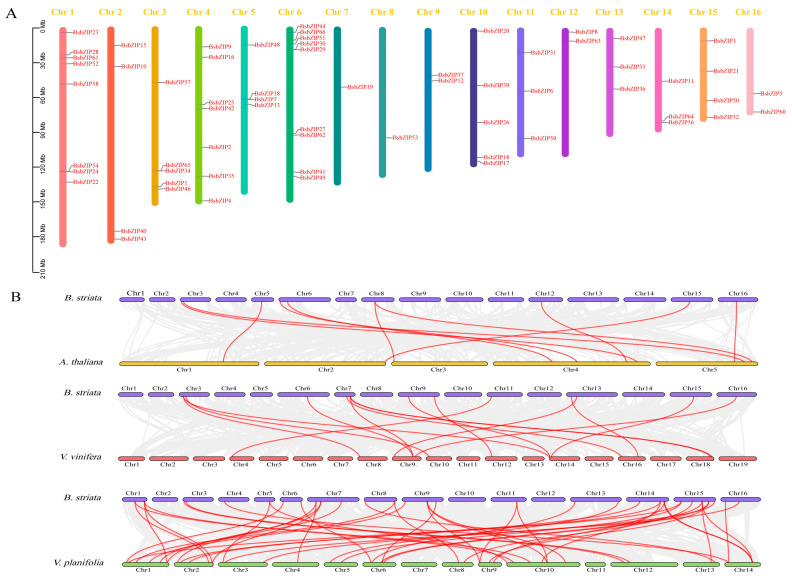
Chromosomal distribution and synteny analysis of *BsbZIP* genes. (**A**) Chromosomal localization of *bZIP* genes in *B. striata*. Three genes (*BsbZIP14, BsbZIP45*, and *BsbZIP55*) were not localized to specific chromosomes. The numbers and sizes of chromosomes are shown at the top and left of each bar chart. (**B**) Synteny analysis of *bZIP* genes between *B. striata* and three other plants. Gray lines represent collinear blocks within the *B. striata* genome and other genomes, and red lines represent bZIP gene pairs.

**Figure 5 ijms-24-15202-f005:**
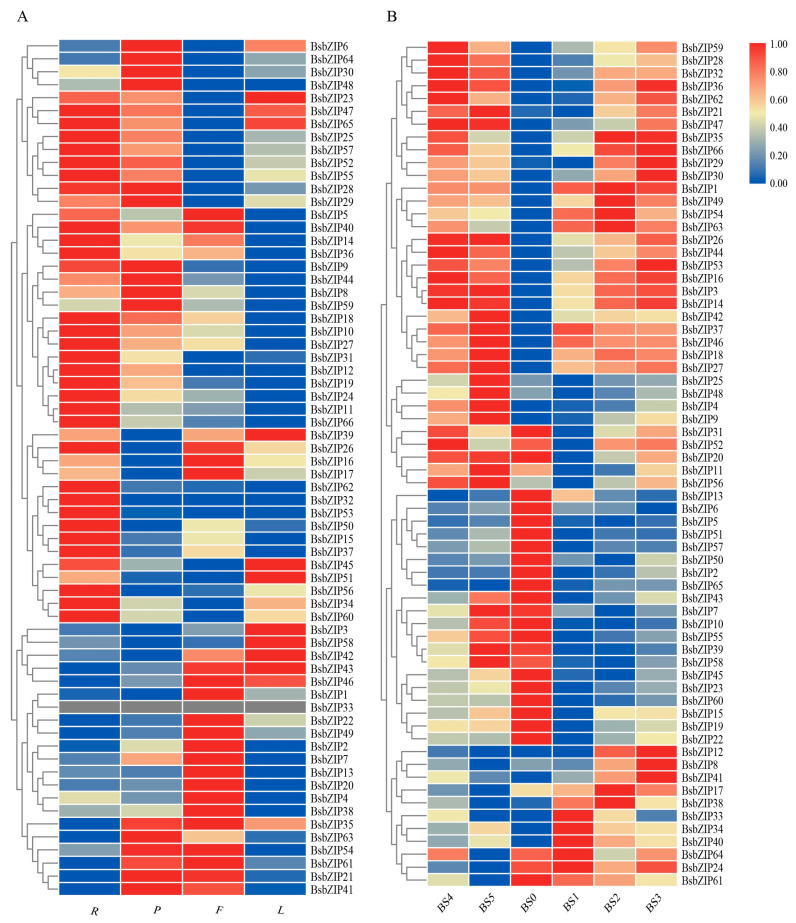
Expression profile heatmap with hierarchal clustering of *BsbZIPs* in various tissues and different germination stages of *B. striata.* (**A**) R, P, L, and F represent roots, pseudobulbs, leaves, and flowers, respectively. (**B**) BS0-BS5 represent different germination stages, respectively: 0 days (BS0), 5 days (BS1), 10 days (BS2), 20 days (BS3), 28 days (BS4), and 35 days (BS5). All data represent averages of three biological replicates. The relative expression for each gene is depicted by the color intensity in each field. Higher values are represented by red, and lower values are represented by blue.

**Figure 6 ijms-24-15202-f006:**
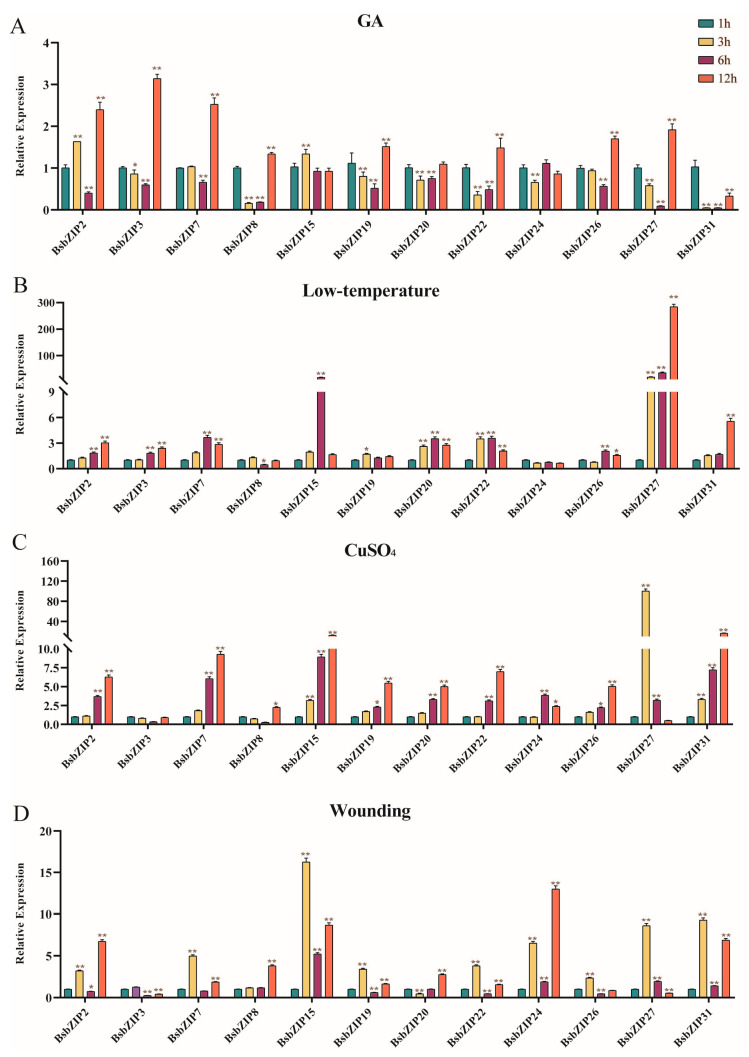
Expression profiles of *BsbZIP* genes under different stress treatments at 1, 3, 6, and 12 h. (**A**) GA. (**B**) Low-temperature. (**C**) CuSO_4_. (**D**) Wounding. All data represent averages of three biological replicates. The data were standardized to the *BsGAPDH* gene and analyzed using GraphPad Prism 8.0. Asterisks (*p* < 0.05) indicate significant differences compared to the controls (* *p* < 0.05, ** *p* < 0.01).

**Figure 7 ijms-24-15202-f007:**
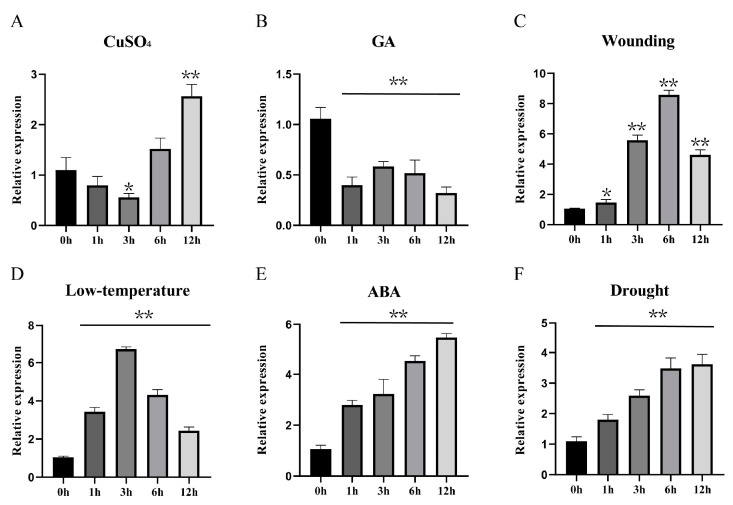
Expression profiles of *BsbZIP13* under different stress treatments. (**A**) CuSO_4_. (**B**) GA. (**C**) Wounding. (**D**) Low-temperature. (**E**) ABA. (**F**) Drought. All data represent averages of three biological replicates. The data were standardized to the *BsGAPDH* gene and analyzed using GraphPad Prism 8.0. Asterisks (*p* < 0.05) indicate significant differences compared to the controls (* *p* < 0.05, ** *p* < 0.01).

**Figure 8 ijms-24-15202-f008:**
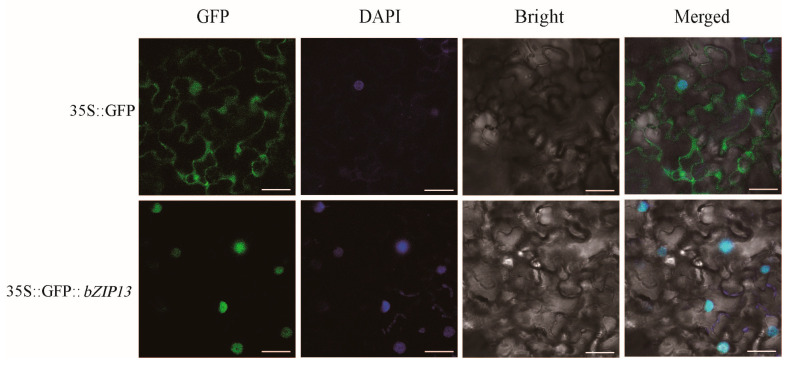
Subcellular localization of *BsbZIP13*. The GFP–BsbZIP13 fusion proteins and GFP as a control were transiently expressed in *N. tabacum* leaves and observed by laser confocal microscopy (Scale bar: 25 µm).

**Figure 9 ijms-24-15202-f009:**
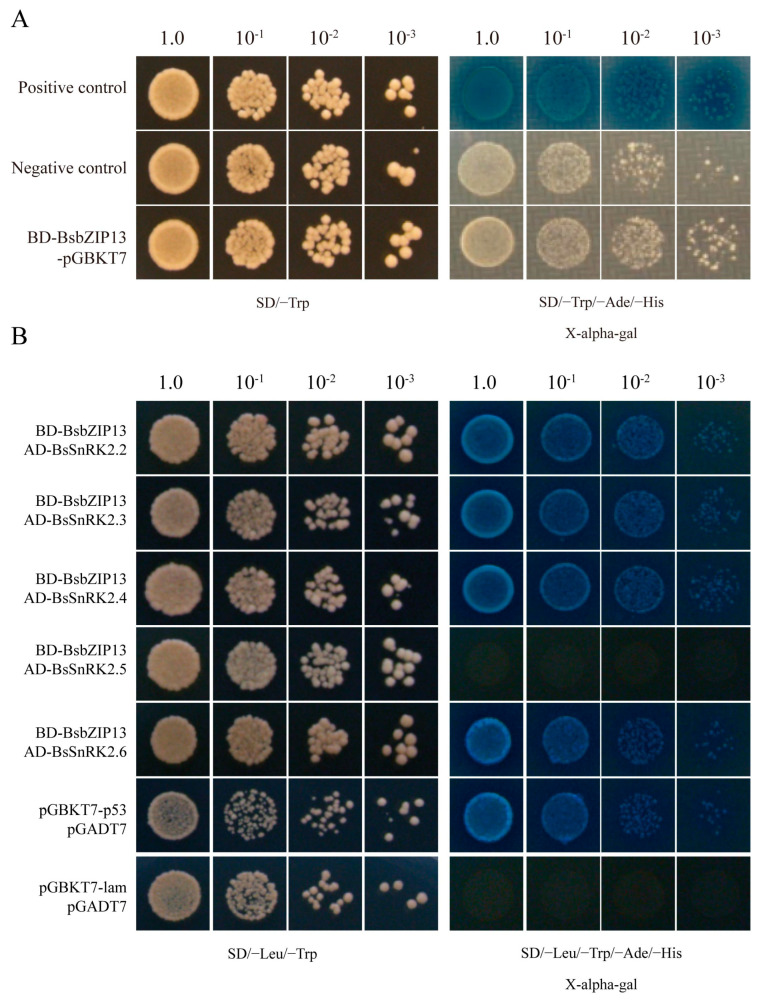
Yeast two-hybrid (Y2H) assays to test the interactions between *BsbZIP13* and *BsSnRK2s*. (**A**) Yeast Transactivation Activity Assay of *BsbZIP13*. Positive constructs, negative constructs, and fusion constructs were transformed into the yeast AH109 strain and successively incubated in SD/−Trp media and a SD−His/−Ade/−Trp plate supplemented with X-alpha-gal. (**B**) Yeast two-hybrid (Y2H) assays to test the interactions between *BsbZIP13* and *BsSnRK2s*. BsbZIP13 was fused with the DNA-binding domain (BD), while BsSnRK2s were fused with the activation domain (AD).

**Figure 10 ijms-24-15202-f010:**
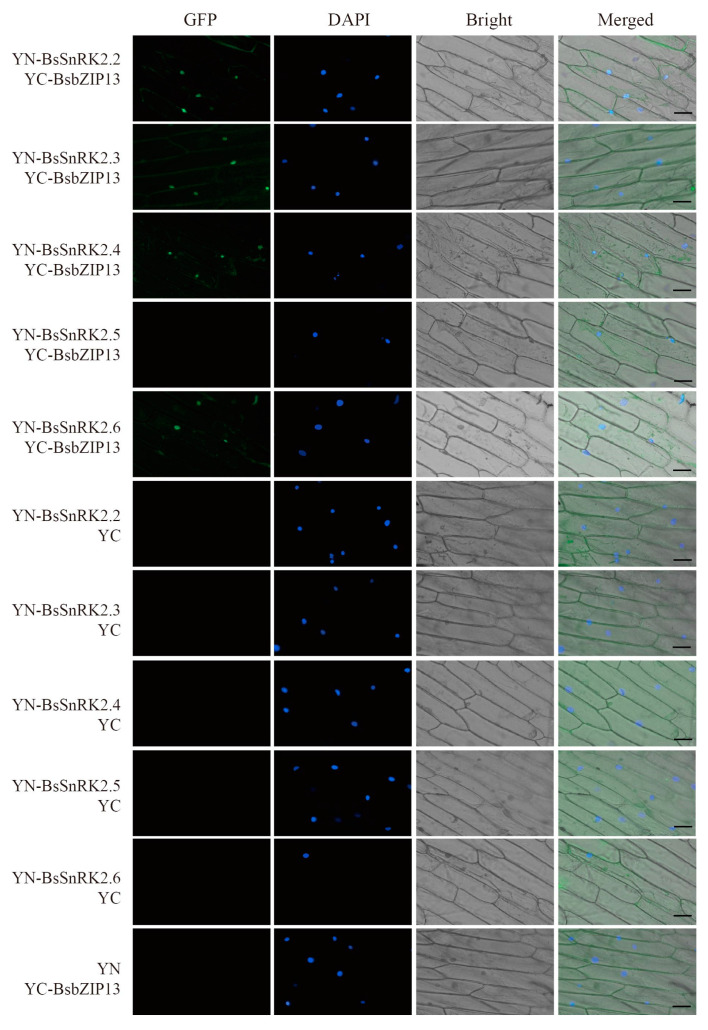
Bimolecular fluorescent complementation (BiFC) experiments in onion epidermal cells to test the interactions between *BsbZIP13* and *BsSnRK2s*. *BsbZIP13* was fused with the C-terminal of fluorescein (YC). BsSnRK2s were fused with the N-terminal of fluorescein (YN). (Scale bar: 75 µm).

## Data Availability

Not applicable.

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
