# Peer review of "Comprehensive Functional Analysis of the bZIP Family in Bletilla striata Reveals That BsbZIP13 Could Respond to Multiple Abiotic Stresses"

_ijms, 2023, doi:10.3390/ijms242015202_

Round 1
Reviewer 1 Report
The study by Zhou et al, provides a comprehensive analysis of the bZIP (basic leucine zipper) transcription factor family in Bletilla striata, a medicinal plant belonging to the Orchidaceae family. The bZIP family is known for its crucial role in regulating gene expression. In B. striata, 66 members of the bZIP gene family were identified, all of which contain conserved structural domains. The study conducted various analyses, including phylogenetic analysis, gene structure, conserved motifs, cis-acting element analysis, followed by synteny analysis, shedding light on the evolutionary relationships and structural features of BsbZIPs. The expression patterns of BsbZIP genes in different tissues and germination stages of B. striata were analyzed using transcriptomic data. The findings highlighted the diverse and essential roles of BsbZIPs in various tissues and stages of plant development. Additionally, the study investigated the response of BsbZIP genes to different abiotic stresses, including hormone treatment, low temperature, heavy metal exposure, and wounding. The results demonstrated the broad involvement of BsbZIPs in abiotic stress tolerance mechanisms. BsbZIP 27 showed very high expression levels with all the treatments. BsbZIP15 showed significant upregulation at different time points with different abiotic stress conditions. BsbZIP13 was responsive to the stimuli of multiple abiotic stresses and showed significant upregulation at all time points under ABA and drought treatments. Finally, the study focused on the functional characterization of BsbZIP13 by creating fusion protein expression vectors, revealing its nuclear localization and interaction with specific SnRK2 proteins (via Y2H and BiFC), which are known to play a role in stress responses.
Overall, this research provides a comprehensive understanding of the bZIP transcription factor family in B. striata, offering valuable insights into their roles in plant growth, development, and response to abiotic stresses. These findings are particularly relevant for the molecular breeding and genetic engineering of B. striata to enhance its resilience to changing environmental conditions.
I highly recommend the publication of this manuscript which will be of interest to the readers of IJMS.
I have a few minor concerns that could be addressed:
1. In Figure 6, the Y-axis label should be 'Relative Expression' instead of 'Relative Expressive'
2. In Figures 6 and 7, the '4' in CuSO4 should be subscripted.
3. In Figure 6, (C) should be CuSO4 instead of GuSO4.
4. In Figure 9, X-a-gal can be written as X-alpha-gal for clarity.
Author Response
Dear Reviewer,
Many thanks for your letter and comments about our paper submitted to International Journal of Molecular Sciences (Manuscript ID: ijms-2621521, Title: Comprehensive functional analysis of the bZIP family in Bletilla striata reveals that BsbZIP13 interacts with BsSnRK2s in response to multiple abiotic stresses). We have carefully revised our manuscript again according to your comments and suggestion. Many details are in the text and some explanations regarding the comments are as follows. (In the revised manuscript, the corresponding revisions regarding to the comments were red marker for Reviewers.)
Please see the attachment.
Response to Reviewer’ Comments
- In Figure 6, the Y-axis label should be 'Relative Expression' instead of 'Relative Expressive'
Answer: Thanks very much for your careful comments and suggestion. We are sorry for the imprecise part of our writing, according to your suggestion, we have revised “Relative Expressive” to “'Relative Expression” in the revised manuscript. (In Figure 6)

Reviewer 2 Report
The paper “Comprehensive functional analysis of the bZIP family in Bletillastriata reveals that Bsbzip13 interact with BsSnRK2s in response to multiple abiotic stresses.” By Ru Zhou et al. report that there are 66 bZIP genes in Bletilla striata and they analyzed gene structure, protein structure, cis-element in promoters, and so on. They found BsbZIP13 responded several abiotic stresses, and it showed interaction with BsSnRK2s. Because the structure of the paper is bad, it is difficult to understand the value of the paper. There are many concerns.
(Because there is no line number, it is hard to point out the problem.)
1. Figure 2. Generally, too small to see. Because B and C show the structure of proteins, the horizontal axis should be N-terminal and C-terminal, not 5’ and 3’, and (a.a) is usually added to the far right of the horizontal axis.
2. Figure 2. What the difference between A and B. it looks B is included in A. For example, bZIP HGP1b-like of BstbZIP 56, 58, and 60 in figure 2C seem corresponding to motif 1 in Figure 2B. it is very confusing. What are the conserved motifs in Figure 2C? In this meaning, motifs in Figure 2B are also conserved among the group. Please clarify the difference. It is better to indicate which motifs are DNA binding motifs.
3. Figure 2D, it is too small to see. yellow bars and light green bars are very difficult to see. these figures should be enlarged, and colors should be changed. Genome size of BstbZIPs are so different and it looked difficult to show in the same figure. It is better to classify them. I think this figure can be in the supplemental data. If we can see the figures, it is no meaning.
This figure shows DNA, (bp) should be added to the far right of the horizontal axis. The genome size of BstbZIP looks more than 80 kbp, and several other BstbZIPs are more than 40 kbp. Is it true? It seems too large.
4. p4 2.4 analysis of cis-acting element: Fig. 3. The analyzed promoter region is 2.5 kb? From transcription start site or translation initiation site? Usually, promoter region is counting from the start site of translation or transcription, and number on the far right should be -1. If the number is base pair, add (bp) in the figure.
5. Figure 5B: the authors should note what are BS0-BS5 in the legend. I cannot understand why the order of the BS0 to BS5 on the bottom of the heatmap is started with BS4.
6.p7. 2.7 section. The authors state that they examined BsbZIP response to abiotic stresses. However, they examined the expression in response to GA and CuSO4. GA is not abiotic stress and we cannot understand why the authors examined the response to CuSO4. The authors should explain it.
Besides, the if the authors want to examine the stress responses, response to ABA should be examined. I cannot understand why they examined just GA.
In the introduction, the authors mention about salinity stress. However, they did not check the response to the salinity.
7. p8. 2.7 section. The authors focus on BsbZIP13. However, there is no BsbIP13 in Figure 6. I cannot understand why the data of BsbZIP13 was separated from Figure 6 and placed in figure 7. The data of BsbZIP13 should be Fig.7.
8. it is unclear why the authors focus on BsbZIP13. In discussion, they state that BsbZIP13 is closest to ABI5. If the authors focus on BsbZIP13 because it may be the ortholog of ABI5, they should explain it in the result section. We cannot understand why the authors start to talk about BsbZIP13.
9. Fig. 7. The authors examined the expression of BsbZIP13 response to ABA and drought. Why they did not examined other genes?
10. Fig.9. The authors examined the interaction of BsbZIP13 with BsSnRK2 without any explanation. We cannot understand way authors examined it. the readers will get lost.
11. p13 L10-20 strange sentence. I cannot understand why there is “whereas” here. The sentence should be reconsidered. The usage of conjunction is wrong.
12. p13 L23-26 the authors mention ABF family, TGA family and so on. however, there is no mention about these families in the phylogenetic analysis(Fig. 1). At least, authors should write which AtbZIP transcription factors are belong to these family. We don’t know the function of these families. The authors should explain it.
13. p14. L14 to L20 What is SlAREB1?
The authors mentioned several genes response to low temperature. However, there is no explanation which BsbZIPs are close to these genes. the authors should mention it (at least which group in Fig. 1 these genes are belonging to).
14. p14 L21-25 the authors should explain why they examined the response to GA.
15. p14 L27 the authors mention here that AtbZIP39 is corresponding to ABI5. The authors should write the name of these famous gene in Fig. 1.
16. p14L35-39 the authors discuss about the transcriptional activity of BsbZIP13 and because it has no transcriptional activity, they need post-translational modification. The transcriptional activity in yeast is different from that in plants. Besides, if the transcription factors have no activity, they can regulate genes by interacting with other transcription factors even though they have not post- translational modification. Therefore, the discussion in not correct.
17. p15. 4.2 the authors state that they used MEGA v7.0. however, in the legend of Fig. 1, they used MEGA-X.
18. p16. L6. There are two BS2
19.p16. L10-20 the used 100 mL GA, and ABA, and 100 mMCuSO4. Why the authors selected the concentration? Is it meaning concentration? Authors should explain why they used the concentration with references.
They sprayed 20 % PEG6000 to induce drought. The authors should show the evidence they can use this method as drought induction.
20. the title: interaction of BsbZIP13 with BsSnRK2s are not induced by abiotic stresses (the authors did not show it). therefore, the title is not correct.
Author Response
Dear Reviewer,
Many thanks for your letter and comments about our paper submitted to International Journal of Molecular Sciences (Manuscript ID: ijms-2621521, Title: Comprehensive functional analysis of the bZIP family in Bletilla striata reveals that BsbZIP13 interacts with BsSnRK2s in response to multiple abiotic stresses). We have carefully revised our manuscript again according to your comments and suggestion. Many details are in the text and some explanations regarding the comments are as follows. (In the revised manuscript, the corresponding revisions regarding to the comments were red marker for Reviewers.)
Please see the attachment.
Response to Reviewer’ Comments
1 Figure 2. Generally, too small to see. Because B and C show the structure of proteins, the horizontal axis should be N-terminal and C-terminal, not 5’ and 3’, and (a.a) is usually added to the far right of the horizontal axis.
Answer: Thank you very much for the professional suggestion. According to your suggestion, we have made changes to Fig. 2. We have revised the horizontal axis“5’ and 3’” to “N-terminal and C-terminal”, and added “(a.a)” to the far right of the horizontal axis of Fig. 2(B).

Reviewer 3 Report
The manuscript by Zhou et al. provides insights into the Basic Leucine Zipper (bZIP) transcription factors (TFs) in Bletilla striata. The study involves the identification and categorization of bZIP family members, structural analysis, chromosomal distribution, expression patterns and protein interactions, and contributes to the understanding of bZIP TFs as regulators of stress responses and their potential implications for transgenic and functional research.
However, there are several critical remarks that demand consideration:
· At first, I would recommend that the authors use the journal template, which includes line numbers and makes it easier to address specific issues.
· The abstract is lacking a proper introduction that contextualizes the significance of the research. Why is studying bZIP TFs in Bletilla striata important? The authors should consider including one to two introductory sentences that elucidate the broader context or problem addressed by this research.
· In the abstract at first introduce the full Latin name of Bletilla striata and then use the abbreviation.
· p. 1, last raw: ‘AtbZIP9/46 protein’ should not be unitalicized as it represents a protein.
· p. 2, line 4 from above: Please rephrase ‘..was able to activate the key enzyme gene the promoters for artemisinin synthesis….’
· p. 3, line 9 from below: Could you clarify the statement ‘Fourteen genes had no introns, accounting for 21.2%’ to provide a more detailed explanation of what the 21.2% refers to?
· p. 4, lines 5-7 from below: The statement ‘Following the removal of general transcriptional regulatory elements and functionally unknown elements, a variety of important cis-acting elements were revealed to be widely distributed in BsbZIPs’ requires supporting results, reference(s) or rephrasing to indicate that such findings will be discussed in a subsequent section of the manuscript.
· p. 5, line 12 from below: ‘…were abundant on (9 genes)’ abundant on what?
· p. 5, last sentence: The sentence "The results indicated that B. striata and V. planifolia possessed higher homologies" is somewhat vague. It is unclear what ‘higher homologies’ refers to without additional context or specific details.
· p. 9, Figure 5 legend: It would be beneficial to include additional information about the different ‘BS stages’. This would save the reader from the need to search for their meaning in the text. Also, consider using a regular hyphen in 'BS0-BS5,' instead of a tilde.
· p.13, line 14 from above: Instead of ‘those report’, use ‘those reports’
· p.13, lines 16-18 from above: Replace ‘Monocotyledon’ and ‘Dicotyledon’ with ‘Monocotyledons’ and ‘Dicotyledons’ for precision.
· p.16: ‘sowed into a solid MS medium with NAA and 6-BA and incubated at 30℃’ - there is no any information about the concentration of NAA and 6-BA.
· p.16, fourth paragraph: Replace ‘grobacterium’ with ‘Agrobacterium’ for accuracy.
· p. 17: Conclusion section must be improved. It lacks a clear, concise summary of the main contributions and their significance. Phrases like "revealed a fundamental understanding" and "provided useful data" are somewhat vague. Instead, specify what exactly was discovered or achieved and provide forward-looking statement, such as suggesting potential areas for future research or applications of the findings.
The manuscript could benefit from proofreading for grammar and language issues to ensure clarity and readability.
Author Response
Dear Reviewer,
Many thanks for your letter and comments about our paper submitted to International Journal of Molecular Sciences (Manuscript ID: ijms-2621521, Title: Comprehensive functional analysis of the bZIP family in Bletilla striata reveals that BsbZIP13 interacts with BsSnRK2s in response to multiple abiotic stresses). We have carefully revised our manuscript again according to your comments and suggestion. Many details are in the text and some explanations regarding the comments are as follows. (In the revised manuscript, the corresponding revisions regarding to the comments were red marker for Reviewers.)
Please see the attachment.
Response to Reviewer’ Comments
- 1, last raw: ‘AtbZIP9/46 protein’ should not be unitalicized as it represents a protein.
Answer: Many thanks for your careful comments. We have revised “ The AtbZIP9/46 protein plays a... ” to “ The AtbZIP9/46 proteins play a...” [p. 1, last raw].
